# On the interplay between lipids and asymmetric dynamics of an NBS degenerate ABC transporter

Ágota Tóth [1], Angelika Janaszkiewicz[1], Veronica Crespi[1] & Florent Di Meo [1✉]

Multidrug resistance-associated proteins are ABC C-family exporters. They are crucial in pharmacology as they transport various substrates across membranes. However, the role of the degenerate nucleotide-binding site (NBS) remains unclear likewise the interplay with the surrounding lipid environment. Here, we propose a dynamic and structural overview of MRP1 from *ca*. 110 μs molecular dynamics simulations. ATP binding to NBS1 is likely maintained along several transport cycles. Asymmetric NBD behaviour is ensured by lower signal transduction from NBD1 to the rest of the protein owing to the absence of ball-and-socket conformation between NBD1 and coupling helices. Even though surrounding lipids play an active role in the allosteric communication between the substrate-binding pocket and NBDs, our results suggest that lipid composition has a limited impact, mostly by affecting transport kinetics. We believe that our work can be extended to other degenerate NBS ABC proteins and provide hints for deciphering mechanistic differences among ABC transporters.

[1] Inserm U1248 Pharmacology & Transplantation, ΩHealth Institute—Univ. Limoges, 2 rue du Prof. Descottes, 87000 F Limoges, France. ✉email: Florent.di-meo@inserm.fr

ATP-binding cassette (ABC) transporters belong to one of the largest trans-kingdom protein superfamilies. The structural resolution of several ABC transporters has led to different conformations (namely, inward-facing—IF, outward-facing—OF, and occluded), which illustrate the alternating access as the most likely model to rationalise substrate translocation along the transport cycle[1–3]. ABC transporter structures are made of at least two transmembrane domains (TMDs) consisting of six transmembrane helices (TMHs). TMDs are bound to two nucleotide-binding domains (NBDs) which are evolutionarily conserved over species. ABC transport cycle requires the binding of two ATP molecules and the energy released from the hydrolysis of at least one of them[3–7]. ATP molecules bind at the interface of the NBD dimer which adopts a non-covalent pseudo-symmetric head-to-tail arrangement; enabling the formation of two nucleotide-binding sites (NBSs). Both NBSs are formed by the conserved Walker A- and B-motifs, the A-, Q- and H-loops of one NBD and the ABC signature sequence and the X-loop of the other NBD[4,5,7,8].

Except for a few members (i.e., CFTR, ABCA4, ABCD4, SUR1/2)[2,3,9,10], eukaryotic ABC transporters are exporters, i.e., they extrude substrates to the extracellular compartment. Eukaryotic ABC transporters used to be classified into type I (ABCB, ABCC, ABCD) and type II (ABCA, ABCG) families. Recently, the structural and functional diversities of ABC transporters have led to a new folding-based classification[2,3] in which the previous type I and type II exporters adopt the type IV and V folding, respectively.

Multidrug resistance-associated proteins (MRPs) are NBS degenerate ABC transporters[3–5,7,8]. In the non-canonical NBS1, the Walker B catalytic glutamate, the A-loop tyrosine, and the first glycine residue of the ABC signature motif are mutated into aspartate, tryptophan, and valine residues, respectively. These mutations were associated with significantly lower ATPase activity and higher ATP-binding affinity for the degenerate NBS1[4,7]. These observations have recently led to the development of a new asymmetric model for NBD function which may affect the dynamics and function of the whole transporter[3,7]. The function and kinetics of bovine ABCC1/MRP1 (bMRP1) were extensively and thoroughly investigated by combining structural information from cryo-electron microscopy (cryo-EM) experiments and single-molecule Förster Resonance Energy Transfer (smFRET)[5]. Despite the robust insights provided by the resolution of bMRP1 structure, unexplained differences between ABCB and ABCC exporters were observed, partially due to the non-native detergent-based environment used for bMRP1 experiments[7,11].

ABC transporters play a crucial role in pharmacology by transporting a tremendous variety of substrates, including xenobiotics and endogenous compounds, across cell membranes. For instance, their pharmacological role has been stressed by the International Transporter Consortium (ITC) which draws out a list of transporters of "emerging clinical importance" for which interactions with new xenobiotics have to be investigated in drug development[12]. Over the past decades, ABCC transporters, in which MRPs are included, have gained a growing interest owing to their role in pharmacology including patient inter-individual responses to treatments. For instance, investigations on ABCC2/MRP2 and ABCC4/MRP4 have been recommended by the ITC to retrospectively provide a mechanistic explanation of clinical observations regarding drug dispositions[13]. Given the role of MRPs in local pharmacokinetics and pharmacodynamics relationships (PK/PD) and therefore in local drug bioavailability[14], there is still a need to decipher in situ MRP transport cycle to provide a comprehensive overview of xenobiotic membrane crossing events. This is particularly relevant for MRPs located in proximal tubular kidney cells and liver hepatocytes since kidneys and liver are involved in the elimination of most worldwide used xenobiotics[15].

Unfortunately, there is no experimentally resolved structure for human MRPs yet. An MD-refined protein threading MRP4 structure[16] has been proposed for which the computational resolution precludes functional investigations or thorough structural dynamics. However, several conformations of bMRP1 transporter have been resolved by cryo-EM[4,5,8]. Given the high sequence similarity to the human ortholog hMRP1 (91%) as well as within other human MRPs (ca. 40–50%), the use of bMRP1 structures as prototype appears relevant for investigating MRP transporter dynamics and functions[3]. MRP1 exporter adopts the type IV folding; however, it has an extra N-terminal TMD made of five TMHs. The so-called $TMD_0$ was shown not to play a role either in ABC ATPase activity or substrate transport[3,4,8]. Therefore, a $TMD_0$-less MRP1 model can be used as a prototype for ABCC exporters even for those which do not possess this domain (e.g., MRP4). $TMD_0$ is connected to conventional ABC TMDs by a linker ($L_0$) which was shown to be mandatory for both trafficking and function[8,17]. The $L_0$ sequence is conserved in all members of the ABCC subfamily even in absence of $TMD_0$[8]. It is important to note that since the present model does not include $TMD_0$, the standard TMH labelling for ABC type IV will be used in the present manuscript, i.e., TMH1 to TMH12.

The present work aims to map the ABC conformational space[1,18] of MRP1 considering different bound states (namely, apo, ATP and/or leukotriene C4—LTX—bound states[8] for IF conformation and ATP bound state[4] for OF conformation) highlighting the importance of asymmetry in ABC domains. Furthermore, given the importance of surrounding lipids in the ABCC transport cycle[11,19], the interplay between the lipid bilayer and protein dynamics was also investigated. This was achieved by using different computational symmetric membrane models made of (i) pure POPC (1-palmitoyl-2-oleoyl-sn-glycero-3-phosphocholine), (ii) pure POPE (1-palmitoyl-2-oleoyl-sn-glycero-3-phosphoethanolamine), (iii) POPC:POPE (3:1), (iv) POPC:Chol (3:1) and (v) POPC:POPE:Chol (2:1:1); the last being the closest to mimic in situ MRP1 dynamics. All-atom unbiased microsecond-scaled molecular dynamics (MD) simulations were conducted to address the objectives.

## Results

**Projection of MRP1 onto the ABC conformational space**. To examine the conformational space sampled during the simulations in POPC:POPE:Chol (2:1:1) accounting for bound states of bMRP1, different structural descriptors were considered according to previous studies[1,11,18]. Namely, intracellular (IC) and extracellular (EC) angles were monitored for TMDs while NBD distance and NBD rocking-twist angle were used for NBDs (Fig. 1 and Supplementary Figs. 1–6); the latter being known to adapt along OF-to-IF transition in ABCB1/P-gp[18]. These structural descriptors were also measured on a large dataset of experimentally resolved ABC proteins including bMRP1 cryo-EM structures[4,5,8] in order to reconstruct the ABC conformational space[1,18] (Fig. 1d and Supplementary Table 1). The known ABC conformations were efficiently featured, i.e., IF open and occluded, OF as well as the recently resolved asymmetric unlock-returned (UR) turnover[1] conformations. bMRP1 MD simulations revealed the spontaneous closing of the intracellular cavity for all IF systems regardless of the bound state and the membrane composition (Fig. 1a, b, Supplementary Figs. 1–6 and Supplementary Tables 2–5). Average NBD distances and IC angles were significantly smaller than those calculated for bMRP1 cryo-EM

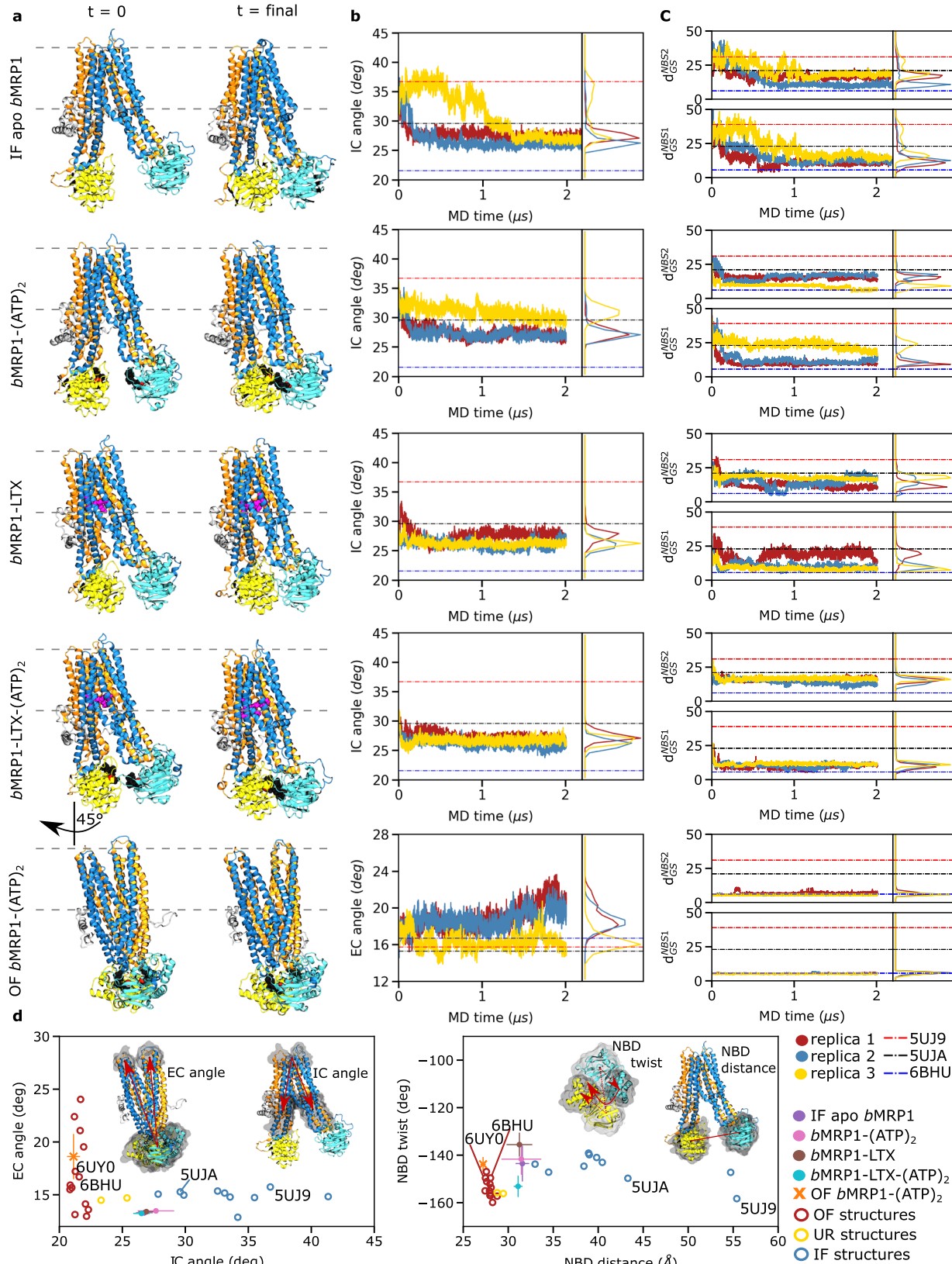

structures. For instance, IF conformations converged toward similar IC angle and NBD distance values ranging from $26.0 \pm 0.5$ to $35.4 \pm 1.8°$ and from $30.3 \pm 1.9$ to $55.0 \pm 2.8$ Å, respectively (Fig. 1b, Supplementary Figs. 1, 3–4, and Supplementary Tables 3–4). Interestingly, the spontaneous dimerisation of NBDs was also observed in absence of ATP molecules. The difference

between simulations and cryo-EM structure might be partially explained by the use of non-native detergent in experiments[7,11]. Indeed, resolved cryo-EM structures of $b$MRP1 exhibited larger IC angles and NBD distances than other ABC structures resolved in native environments (Fig. 1d). On the extracellular side of the lipid bilayer membrane, MD simulations of OF $b$MRP1-(ATP)₂

**Fig. 1 Overview of _b_MRP1 milestone structures along the transport cycle in POPC:POPE:Chol (2:1:1).** _Inward-facing conformations:_ apo-state (IF apo _b_MRP1), substrate-bound (_b_MRP1-LTX), ATP-bound (_b_MRP1-(ATP)$_2$), ATP/substrate-bound (_b_MRP1-LTX-(ATP)$_2$); _Outward-facing conformation:_ ATP-bound (OF _b_MRP1-(ATP)$_2$). **a** Representative snapshots of the different _b_MRP1 systems investigated here at the beginning and the end of MD simulations. **b** Time-evolution of the IC and EC angles respectively for IF and OF structures. IC and EC angles were calculated according to the proposed ABC structural parameters defined in the Methods section[1,11,18]. **c** Time-evolution of key NBS distances defined by inter-NBD distances between Walker A glycine and ABC signature serine residues[8]. **d** Projection of _b_MRP1 structural parameters onto the ABC conformational space obtained from multiple resolved ABC structures. Results were obtained from _n_ = 3 MD trajectories for each system and for which standard deviations are shown. PDB IDs of the resolved _b_MRP1 cryo-EM structures are explicitly mentioned. The first and second TMDs are respectively depicted in orange and blue, and NBD1 and NBD2 are respectively coloured yellow and cyan.

displayed minor openings of the EC gate as compared to _b_MRP1 OF cryo-EM structures, suggesting the existence of a slightly more open state. However, calculated EC angles remained small (lower than 20° in POPC:POPE:Chol (2:1:1), see Fig. 1a and Supplementary Figs. 1–2 and Supplementary Table 2), which precludes the substrate re-entry[1].

Even though trajectories performed with the pre-translocation state (i.e., _b_MRP1-LTX-(ATP)$_2$) tend to populate the ABC conformational subspace of OF conformation (Fig. 1d), substrate translocation was not observed. NBD twist values calculated for _b_MRP1-LTX-(ATP)$_2$ state are similar to OF ABC conformations. However, NBD distances remain larger than for resolved substrate-free OF ABC structures and OF _b_MRP1-(ATP)$_2$ simulations. Distances between Cα atoms of the ABC signature motif serine and a Walker A glycine were monitored (i.e., Gly681-Ser1430 and Ser769-Gly1329, respectively denoted as $d_{GS}^{NBS1}$ and $d_{GS}^{NBS2}$, Fig. 1c, Supplementary Figs. 7–8). Structural differences between the pre- and the post-translocation states (i.e., _b_MRP1-LTX-(ATP)$_2$ and OF _b_MRP1-(ATP)$_2$, respectively) suggest that a conformational transition of NBD dimer is required prior to the substrate translocation event. Such transition from the so-called "non-competent" to "competent" NBD dimer conformations is likely to trigger TMD conformational transitions suggesting that it might be the limiting step for IF-to-OF transition.

_b_MRP1 conformations can also be documented on the basis of TM pore opening (Supplementary Figs. 9–13). As expected, TM pore is larger for OF _b_MRP1-(ATP)$_2$ in the upper leaflet (at z = 18 and 5 Å above lipid bilayer centre, Supplementary Figs. 10–11) while IF conformations remain closed. Interestingly, a bottleneck shape was observed for IF conformation at 5 Å. In spite of the dynamic variability observed for TM radius profiles in the lower leaflet, the following sequence in term of opening was observed (Supplementary Figs. 12–13): IF apo _b_MRP1 > IF _b_MRP1-(ATP)$_2$ > IF _b_MRP1-(LTX) ≈ IF _b_MRP1-LTX-(ATP)$_2$ > OF _b_MRP1-(ATP)$_2$. This is in line with the role of substrate which was shown to bind TM bundles in ABC transporters[18], including _b_MRP1[4,5]. In contrast to IC angle and NBD distance which tend to rapidly decrease in early stage of MD simulations, TM pore radii at selected depths in lipid bilayer remain relative constant along simulations.

**Asymmetric dynamics of _b_MRP1 and modulation of its conformational landscape.** Overall flexibilities were assessed by calculating root-mean-square fluctuations (RMSF, Supplementary Fig. 14). Per-residue RMSF confirms the asymmetric behaviour of NBDs[7], NBD2 being more flexible than NBD1. For each system, backbone-based principal component analyses (PCA) were conducted. Only the three first largest principal components were considered, revealing 85.6% to 95.9% of the overall structural variabilities depending on the system (Supplementary Fig. 15). For all IF simulations, the three first largest variabilities were systematically assigned to asymmetric NBD motions for which NBD2 contributed the most, from 27.0 to 59.6% of the motion

(Supplementary Fig. 16). The first principal components were mostly assigned to NBD twist and rocking motions (Supplementary Movies 1–5). This is in agreement with the experimentally suggested higher flexibility of NBDs from smFRET experiments along the kinetic cycle of MRP1[5]. Regarding OF simulations, the two first principal components were associated with concerted NBD twist motion and opening of the extracellular side mediated mostly by TMH4, TMH5, TMH7 and TMH8. These TMHs have been suggested to behave as a single bundle in ABCB1/P-gp[18]. Furthermore, to a lesser extent than for IF conformations, NBD2 remained more involved in this shared motion than NBD1.

The asymmetric behaviour was also pictured by the supramolecular arrangement between NBD1 and NBD2 for which two main subpopulations were observed for apo, _b_MRP1-(ATP)$_2$ and _b_MRP1-LTX states (Fig. 2a). Interestingly, interactions within NBSs across NBDs were maintained along our MD simulations, even in absence of ATP molecules. Two NBD conformations were observed, either asymmetrically open or closed NBD dimer arrangements, in favour of the latter. Surprisingly, both arrangements were also observed but to a lesser extent in _b_MRP1-(ATP)$_2$ even though ATP molecules were expected to maintain interactions at the NBD dimer interface. In presence of both substrate and ATP molecules, only the closed population was observed picturing the information transduction from the TMD substrate-binding pocket to NBDs in order to likely decrease the energy barrier for IF-to-OF transition in type IV folding ABC transporters[20,21].

In order to explain the asymmetric behaviour of NBDs, particular attention was paid to coupling helices (CH) which ensure the signal transduction from TMDs to NBDs[20,21]. Natively, ABC transporters exhibit four coupling helices linking intracellular domains of TMH2/TMH3, TMH4/TMH5, TMH8/TMH9, and TMH10/TMH11. The so-called CH$_{2-3}$ and CH$_{10-11}$ are in contact with NBD1 while CH$_{4-5}$ and CH$_{8-9}$ interact with NBD2 (Fig. 2a). Contacts between CHs and NBDs are not modified upon binding of ATP molecules or substrate (Supplementary Fig. 17). Interestingly, for a given NBD, one can consider a so-called "weak CH" (namely, CH$_{2-3}$ and CH$_{8-9}$ for NBD1 and NBD2, respectively) for which only a few contacts were observed. On the other hand, the so-called "strong CH" (namely, CH$_{10-11}$ and CH$_{4-5}$ for NBD1 and NBD2, respectively) exhibited more contact with the NBD and for which the "ball-and-socket" arrangement is significantly more pronounced than for the "weak CH". Interestingly, in mammalian MRP1, the NBD1 sequence exhibits a 13-aminoacid deletion which is associated with fewer contacts and significantly weaker interactions between CH$_{2-3}$ and CH$_{10-11}$ as compared to CH$_{4-5}$ and CH$_{8-9}$ for NBD2 (Supplementary Fig. 18). Furthermore, the few contacts observed between CH$_{2-3}$ and CH$_{10-11}$ are slightly decreased in presence of ATP molecules while the contact pattern for NBD2, i.e., CH$_{4-5}$ and CH$_{8-9}$, is well conserved regardless of the presence of ATP and/or leukotriene C4. This leads to the disruption of the so-called "ball-and-socket" arrangements of CH$_{2-3}$ and CH$_{10-11}$ in

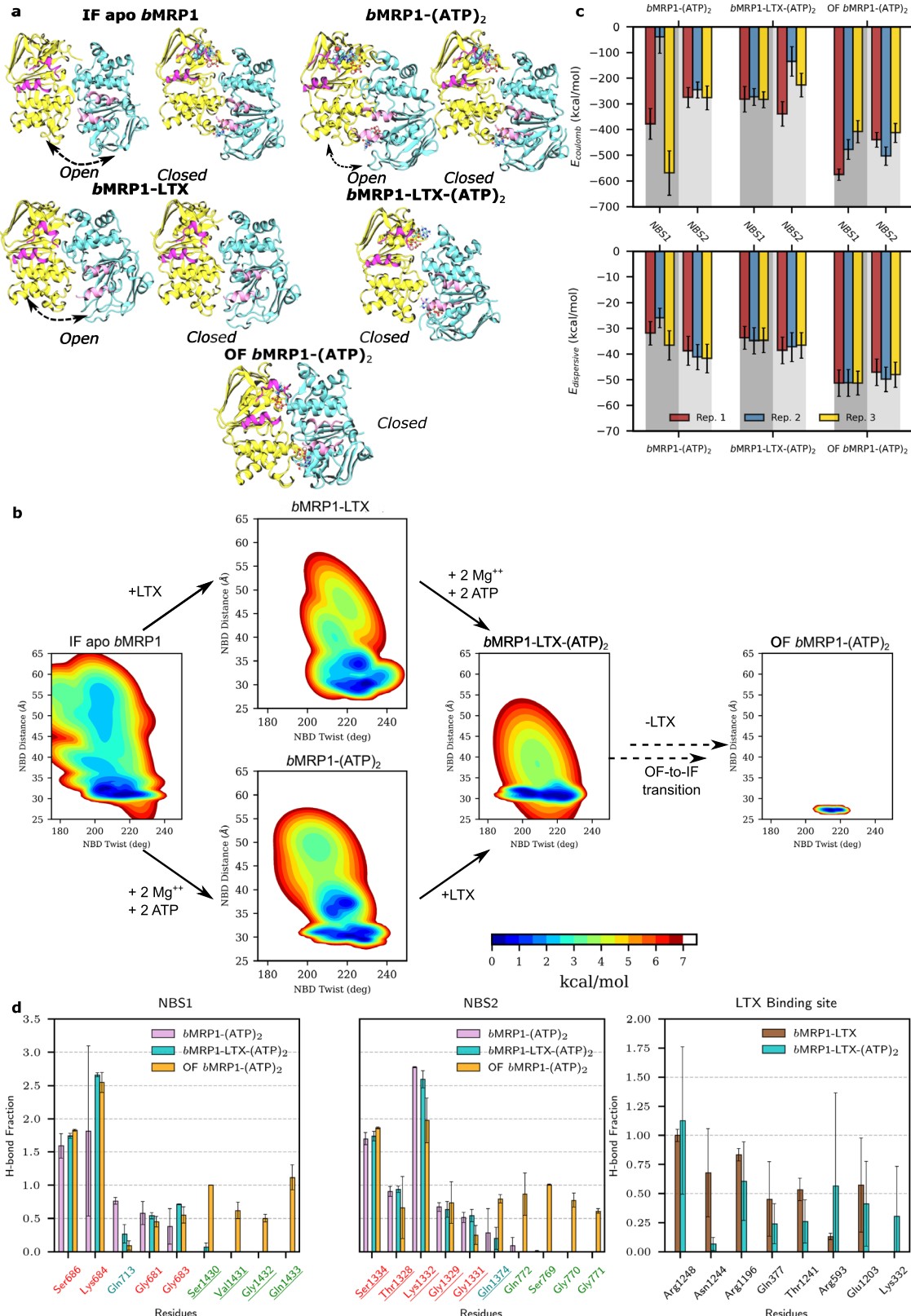

NBD1 which is expected (i) to be responsible for the aforementioned NBD asymmetric opening and (ii) to preclude the information transduction between TMDs and NBD1[8].

IF conformations exhibited weaker NBD dimer interactions than OF conformation, explaining the overall larger flexibility. Likewise, MD simulations revealed slightly higher flexibility of IF

apo state as compared to ATP- and/or LTX-bound systems. This is in agreement with previous observations suggesting that interactions between TMHs or between NBDs are modulated by the presence of substrate and/or ATP molecules[4,7,8,22]. Taking advantage of our extensive unbiased MD simulations, explored conformational subspaces were featured in terms of free energy

**Fig. 2 Asymmetric structural dynamics of *b*MRP1 systems in POPC:POPE:Chol (2:1:1) and their conformational landscape. a** Representative snapshots picturing the open and closed conformations of NBD dimers observed during MD simulations. IF apo *b*MRP1, *b*MRP1-(ATP)$_2$ and *b*MRP1-LTX revealed two main subpopulations for which black arrows highlight the motion; while the pre- and post-translocation conformations (namely, *b*MRP1-LTX-(ATP)$_2$ and OF *b*MRP1-(ATP)$_2$) exhibited only the closed NBD dimer conformation. NBD1, NBD2 and coupling helices are respectively depicted in yellow, cyan and pink. **b** System-dependent local conformational landscapes obtained from the GMM-based approach developed in the InfleCS method[23,64] highlighting the influence of nucleotides and the substrate on *b*MRP1 structural dynamics. **c** Averaged Coulomb and van der Waals potentials calculated between nucleotides and NBS1 and NBS2, separately (800 points were used for each replica, treated independently; error bars show standard deviations). **d** Calculated H-bond networks between ATP and NBS1 or NBS2 and between LTX and substrate-binding pocket residues (H-bond fractions were calculated from each replica independently and were then averaged $n = 3$; error bars showing standard deviation over them).

using the InfleCS clustering methods[23] (Fig. 2b). Given the aforementioned non-competent NBD dimer conformations, focus was paid to NBD structural parameters, namely NBD twist and NBD distance. Larger variabilities for IF conformations were observed leading to multiple plausible minima for which interconversion is possible but slow. The expected to be competent NBD twist *versus* NBD distance subspace tends to be populated in presence of ATP molecule and/or substrate. Such finding highlights the central role of ATP- and substrate-binding events in the ABCC1 transport cycle. To better decipher the NBD dimer dynamics, structural networks[24] were obtained by considering both contact maps (Supplementary Fig. 19) and dynamic cross-correlation matrices (Supplementary Fig. 20). Each NBD can be globally split into two subdomains regardless of the conformation or the bound states driven by ATP-binding (Supplementary Fig. 21). Indeed, similar so-called communities were relatively well-conserved over the replicas and the systems investigated (Supplementary Table 6). The first community is defined by Walker A and B as well as A- and H-loops. On the other hand, the second community includes Q- and X-loops and the ABC signature sequence. Residues involved in these communities are highly correlated suggesting that local residue displacement upon ATP-binding will propagate. The two communities may be considered almost independent; thus, the binding of one ATP is not expected to strongly impact the motions of the second community. Meanwhile, given the asymmetric NBD dimer arrangement, ATP and magnesium are expected to link communities across NBDs. Interestingly, dynamic correlations in IF apo simulations exhibited larger variabilities since communities could be split into subcommunities (Supplementary Table 7), so that ATP-binding is expected to strengthen the structural cooperation within and between NBDs.

**Conformation-dependent binding modes of ATP in degenerate and canonical NBSs.** Particular attention was paid to the binding modes of ATPs by assessing van der Waals and Coulomb potentials as well as H-bond networks in the NBSs (Fig. 2c, d). Such analyses should not be considered quantitatively owing to the compensation of errors between the two potentials, especially at short distances[25]. However, they can be used to provide qualitative hints in order to compare ATP-binding driving forces. Interestingly, both Coulomb and dispersive interactions exhibited less attractive energies for all IF than OF conformations. Given that NBDs exhibit lower flexibility in OF conformation, the subsequent tighter NBD dimer interactions can be rationalised by the proper local arrangement of ATP molecules in NBSs. For instance, calculated lower π-stacking distances between ATP molecules and NBS conserved motifs were systematically larger in IF than in OF conformations leading to lower interaction energies between ATP molecules and MRP1 residues (Fig. 2c and Supplementary Table 8). Surprisingly, in IF conformations, NBS1 tends to exhibit lower dispersive interactions between ATP molecule and Trp653 A-loop residue as compared to Tyr1301 in

NBS2 A-loop, while the spatial aromatic surface of tryptophane is larger than for tyrosine. On the other hand, in the OF conformation, dispersive contributions tend to be slightly larger for NBS1 than for NBS2. H-bond networks between ATP molecules and NBSs (Fig. 2d) suggest a similar network between NBSs for IF conformations. The H-bond network is conserved over IF and OF conformations regarding interactions with Walker A; however, ATP-bound IF systems do not exhibit the expected H-bond network with the signature motif as observed in OF simulations. Interestingly, MD simulations suggest an asymmetric behaviour between NBS regarding H-bonds with Q-loop glutamine residues, namely Gln713 and Gln1374, respectively for NBS1 and NBS2. Calculated distances suggested that ATP γ-phosphate binding to NBS1 Q-loop was weaker than for NBS2 in the OF conformation. The opposite behaviour was observed in MD simulations for IF conformations. Therefore, present simulations underline that proper ATP-binding modes in both NBSs are key in triggering conformational changes required for substrate translocation.

**Towards deciphering the allosteric modulation between substrate- and nucleotide-binding sites.** Particular attention was paid to the substrate-binding pocket and it was compared to the cryo-EM structure of *b*MRP1 bound to leukotriene C4[8]. In agreement with experiments, leukotriene C4 binding mode took place in the two so-called P- and H-pockets ("P" and "H" respectively standing for polar and hydrophobic). Given the amphiphilic feature of leukotriene C4, Coulomb and H-bond networks have been shown to be central for substrate-binding[8]. MD simulations highlighted the same key residues which were experimentally observed (Fig. 2d). For instance, strong salt-bridges were observed between arginine residues (Arg1248, Arg1196 and Arg593, Figs. 2d and 3a) maintaining at least two out of the three leukotriene carboxylate groups in the P-pocket. Interestingly, variabilities in terms of H-bond fractions or inter-action energies (Fig. 2d and Supplementary Fig. 22) suggest a dynamic binding mode in agreement with its expected breaking along the IF-to-OF large-scale conformational changes.

Even though differences in terms of interaction energies and H-bond networks between *b*MRP1-LTX-(ATP)$_2$ and *b*MRP1-LTX or *b*MRP1-(ATP)$_2$ remained low, they suggest a distant effect between NBSs and the substrate-binding pocket. Allosteric effect was assessed between the substrate-binding pocket and NBS1 and NBS2, independently (Fig. 3b). This was achieved by considering key residues for each binding site (Supplementary Table 9) for allosteric pathway network analyses[26,27]. Efficiencies (Fig. 3c) were calculated in presence or absence of substrate and ATP molecules. The impact of POPC:POPE:Chol (2:1:1) lipid bilayer was also considered. Natively, substrate-binding pocket and NBSs are allosterically connected through the protein as shown by calculated efficiencies without including nucleotides or substrates. As expected, the presence of substrate and/or ATP molecules substantially increased the allosteric communication from substrate-binding pocket to both NBSs. Interestingly, in spite of the aforementioned NBD asymmetric dynamics, present

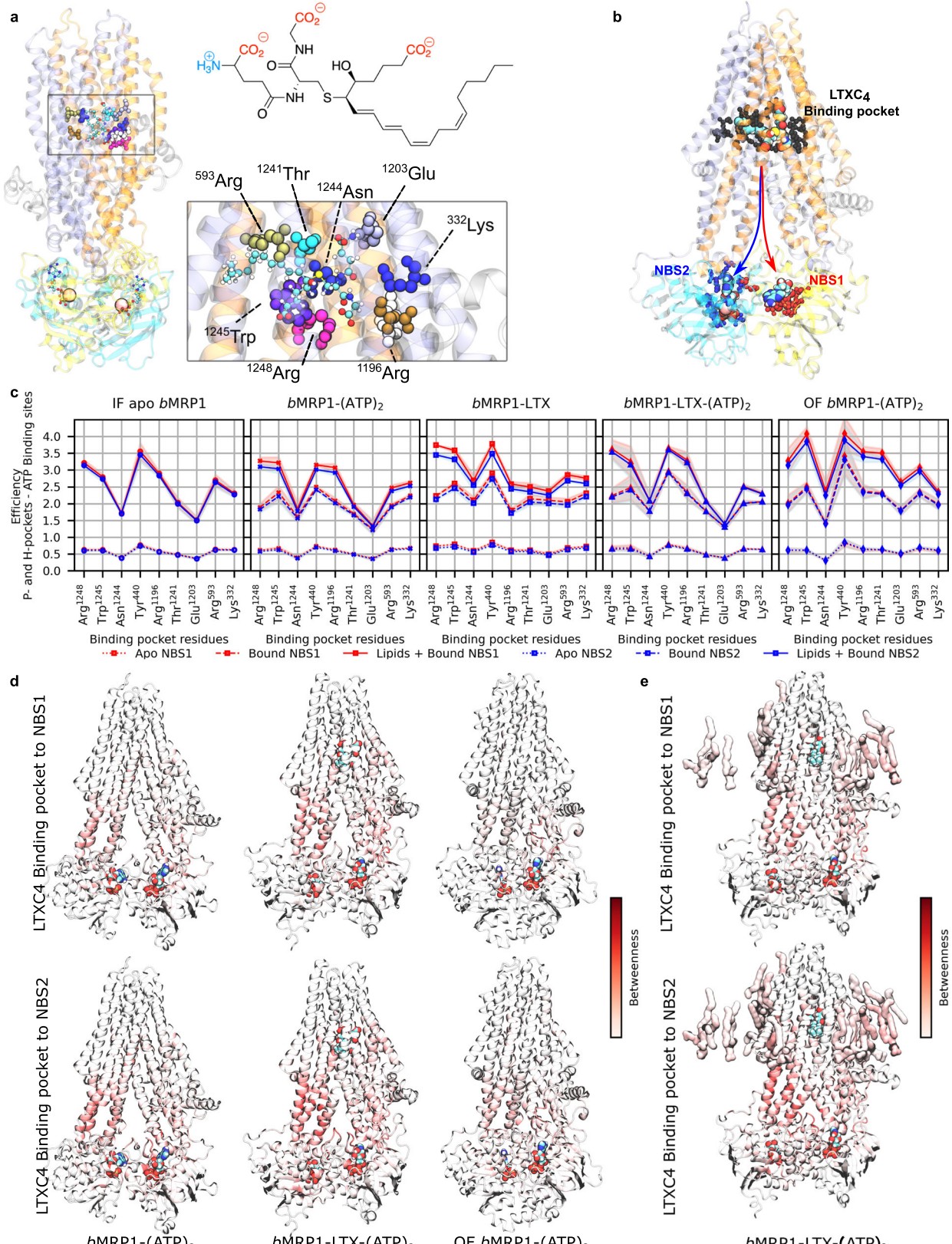

calculations did not exhibit a significant difference in terms of efficiencies between NBSs. Betweennesses were calculated to picture the residue and domain contributions to the allosteric pathway (Fig. 3d and Supplementary Figs. 23–25). Particular attention was paid to ATP-bound systems, i.e., IF $b$MRP1-$(ATP)_2$, and $b$MRP1-LTX-$(ATP)_2$ as well as OF $b$MRP1-$(ATP)_2$

(Fig. 4d), other systems are reported in Supplementary Fig. 26. The main residues involved in the binding pocket-NBS allosteric pathways are mostly located in the intracellular part of TMHs. Interestingly, an asymmetric behaviour, for which TMH4, TMH5, TMH7 and TMH8 are significantly more involved than TMH1, TMH2, TMH10 and TMH11, was again observed. $b$MRP1

**Fig. 3 Substrate-*b*MRP1 interactions and subsequent allosteric communications to nucleotide-binding sites. a** Substrate-binding pocket highlights important residues to leukotriene C4 binding. The structure of leukotriene is also shown for which amphiphilic features are stressed out. **b** Definition of the allosteric pathway investigated in the present study for which NBS1 and NBS2 were treated separately. **c** Calculated allosteric efficiencies of the information flow between substrate-binding pocket and NBS1 (red) or NBS2 (blue) for the different systems embedded in POPC:POPE:Chol (2:1:1). Solid, dashed and dotted lines respectively depict efficiencies considering: Protein + lipids + nucleotides/substrate, Protein + nucleotides/substrate and standalone Protein. Standard errors are shown as shades and were calculated from $n = 3$ replicas treated independently for each system. **d** Protein and **e** lipid contributions to the information flow for allosteric communication from substrate-binding pocket to NBS1 and NBS2 show that NBD2 and its coupling helices (CH$_{4-5}$ and CH$_{8-9}$) are systematically involved regardless of the sink region.

exhibits an asymmetric feature regarding the so-called "ball-and-socket" arrangements which was shown to be responsible for the structural cooperation between TMHs and NBDs through coupling helices[8]. The thirteen amino acid deletion in NBD1 leads to lower cooperation between TMH1, TMH2, TMH10 and TMH11 with NBD1, which in turn decreases the allosteric communication between substrate-binding pocket and NBS1. Furthermore, our calculations suggest that information mostly goes through NBD2 since the direct communication with NBD1 is significantly weakened by the absence of "ball-and-socket" conformation for CH$_{10-11}$[8]. Interestingly, POPC:POPE:Chol lipid bilayer was also shown to play a key role in the allosteric communication between substrate-binding pocket and NBSs (Fig. 3c). However, even though its impact is significant, lipid bilayer contributions appeared milder than in e.g., Major Facilitator Superfamily membrane transporters[28].

**On the interplay between the lipid bilayer and *b*MRP1 structures and dynamics.** As nowadays more attention has been paid to the interplay between the surrounding lipid environment[11,19,22,26,28] and membrane proteins, lipid-dependent protein dynamics and lipid-protein interactions were investigated. This was achieved by carrying out MD simulations in different lipid bilayers, namely POPC, POPC:Chol (3:1), POPC:POPE:Chol (2:1:1). IF apo *b*MRP1 and OF *b*MRP1-(ATP)$_2$ systems were also considered in unrealistic POPE and POPC:POPE (3:1) lipid bilayers.

Projection of lipid-dependent structural parameters onto the ABC conformational space (Supplementary Figs. 1, 27 and 28) revealed that most of the present simulations tend to explore similar ABC subspaces regardless of the lipid bilayer membrane composition. Figure 4a compares structural parameter averages according to lipid bilayer compositions for all MD simulations performed in the present study. For IF conformations, only intracellular structural parameters (i.e., IC angle and NBD distance) exhibited slight deviations according to the lipid composition. Systems performed in pure POPC lipid bilayer exhibited slightly more open conformations. On the other hand, our calculations suggest that only EC angle is affected by lipid bilayer composition but only for OF conformations. Likewise, calculated cavity radii (Supplementary Fig. 9) exhibited small differences while comparing lipid bilayer compositions. Even if these calculations underline a relatively limited overall impact of membrane composition while comparing *b*MRP1 structures in different lipid bilayer models, they did not sufficiently picture the dynamic variability over MD simulations and replicas (Supplementary Figs. 1–6 and 9).

Lipid-dependent conformational landscapes were calculated (Supplementary Figs. 29 and 30). For a given conformation and bound state, MD simulations preferentially populated similar regions regardless of lipid composition. However, structural variability was shown to be significantly affected by lipid composition. For instance, in pure POPC lipid bilayer, more open IF conformation subspaces, ranging from 30 to 55 Å NBD distance, were sampled. This effect was however reduced in

presence of substrate which is expected to tend to maintain more contacts between TMHs and thus reduce the intracellular opening. MD simulations performed on POPC:POPE:Chol (2:1:1) exhibited a significantly smaller sampled region suggesting that the presence of PE lipids tends to close the intracellular gate of *b*MRP1 IF conformations, regardless of the bound state. On the other hand, regarding OF conformations and extracellular opening, global minima were interestingly observed in the same subspace of the conformational space as MD simulations performed in pure POPC and POPC:Chol (3:1). It suggests a limited impact of cholesterol on the opening of *b*MRP1 EC gate. However, the presence of PE lipids slightly shifted the calculated minima toward more opened OF structures, from 13.9 to 18.5°. Tilt angles between TMHs and lipid bilayer normal were measured in order to unravel slight differences between the distinct lipid bilayer compositions (Supplementary Fig. 31). Even though no clear conclusions can be drawn from these analyses, orientations of TMH3, TMH6 and TMH9 appeared more sensitive in absence of cholesterol. While considering that TMHs act as bundles as shown for ABCB1/P-gp[18], the impact of lipid bilayer membrane was more pronounced on the tilt orientations of given smaller bundles (Supplementary Fig. 32), namely Bundle C & D respectively consisting in TMH3/TMH6 and TMH9/TMH12. Interestingly, these bundles are expected to undergo larger conformational changes along the transport cycle[18]. This suggests that even though lipid composition seems to have a rather limited impact while comparing structures of *b*MRP1 local minima in different lipid bilayer membranes, lipid composition is expected to affect conformational transitions, and thus, in turn, play a role in the kinetics of substrate transport by *b*MRP1.

To better understand the interplay between the lipid bilayer and *b*MRP1, particular attention was paid to the lipid bilayer membrane structure. The larger structural variability in pure POPC membrane was explained by a significantly more fluid lipid bilayer structure pictured by lower order parameters ($S_{CD}$) for palmitate and oleate tails (Fig. 4b). In line with the biophysics of pure lipid bilayers, the presence of cholesterol modulates the fluidity of POPC by increasing the lipid order which in turn led to lower flexibility of the lipid bilayer membrane[29]. To a lesser extent, the presence of PE lipid potentiated the structural effect of cholesterol, which is in agreement with the slightly lower structural variability of *b*MRP1 in POPC:POPE:Chol (2:1:1) as compared to other lipid bilayer membranes. Lipid order calculations also suggest a weak impact of protein dynamics on lipid bilayer structures. Indeed, IF apo and OF *b*MRP1-(ATP)$_2$ MD simulations exhibited slightly less ordered lipid tail profiles than IF *b*MRP1-LTX, *b*MRP1-(ATP)$_2$ and *b*MRP1-LTX-(ATP)$_2$. This may be explained by larger variability of intracellular and extracellular openings for IF and OF conformation, respectively, which in turn is likely to lead to more pronounced displacements of surrounding lipids. Membrane free energy deformations[30] were also assessed to document the impact of *b*MRP1 conformations onto lipid bilayer structures (Supplementary Tables 10–12 and Fig. 4c). While calculations suggest that the presence of *b*MRP1 destabilise pure POPC lipid bilayer structure, the

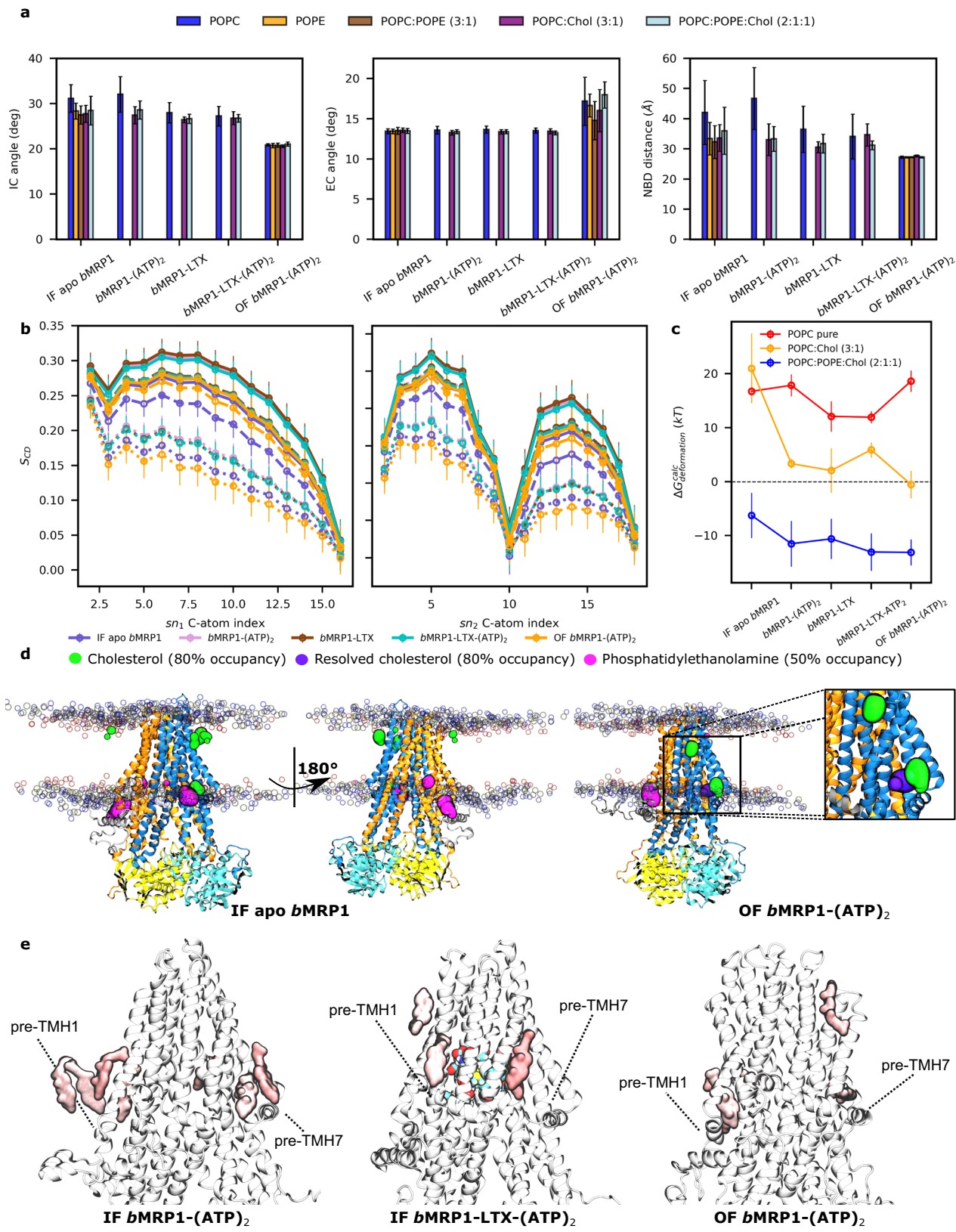

opposite trend was interestingly observed in POPC:POPE:Chol (2:1:1) in which membrane is stabilised in presence of $b$MRP1 ($\Delta G_{deformation} < 0$, see Fig. 4c). An intermediate behaviour was observed in POPC:Chol (3:1) lipid bilayer membrane for which the presence of $b$MRP1 globally destabilised lipid bilayer structure but to a significantly lesser extent than in pure POPC. Except for

simulations performed in pure POPC lipid bilayer, calculated deformation free energies are larger for IF apo $b$MRP1 state than for other conformations, likely due to its aforementioned larger flexibility in absence of ATP and/or substrate.

Lipid-dependent two-dimensional density analyses as well as the assessment of distribution of surrounding lipids revealed

**Fig. 4 Interplay between *b*MRP1 structural dynamics and lipid bilayer according to its composition. a** Average IC angle, EC angle and NBD distance for all *b*MRP1 systems embedded in different lipid bilayers calculated from $n = 3$ MD trajectories per state and lipid bilayer. Error bar refers to as standard deviations of each dataset. **b** Calculated C-atom lipid tail order parameters ($S_{CD}$) for palmitic ($sn_1$) and oleic tails ($sn_2$) for all systems embedded in POPC-based models. Solid, dashed and dotted lines respectively depict lipid order parameters obtained in POPC:POPE:Chol (2:1:1), POPC:Chol (3:1) and POPC lipid bilayer models. Error bar refers to as standard deviations of each calculated dataset. **c** Calculated membrane free energy deformations[30] obtained by averaging over $n = 3$ MD trajectories, considering each replica independently. Raw data are available in Supplementary Tables 10–12. Error bars refers to as standard deviation over the three independent replicas. **d** Calculated binding hotspots obtained from cholesterol and PE lipids defined by presence likelihood higher than 80 and 50% respectively for cholesterol and PE lipids. Zoom on Cryo-EM resolved cholesterol (violet) is shown to highlight the specific parallel orientation with respect to the lipid bilayer. **e** Important lipid areas based on allosteric pathway analysis. The location of key cholesterol molecules involved in the allosteric communication between substrate-binding pocket and NBSs is shown. Both NBSs are considered, highlighting the expected central roles of $L_0$, pre-TMH1 and pre-TMH7 regions in specific lipid-protein interactions which might favour *b*MRP1 function.

important cholesterol and PE lipid hotspots. For instance, PE lipids were shown to preferentially bind to pre-TMH7 elbow helix as well as close to the $L_0$ domain, for more than 50% of the simulation (Fig. 4d, calculated 2D density profiles in Supplementary Figs. 33–36). Electron density maps revealed three cholesterol molecules bound to the resolved OF *b*MRP1 structure from which two was maintained near to its initial position with a probability higher than 50% (Supplementary Fig. 33). Interestingly, one by the pre-TMH7 is strongly (more than 80%) maintained along the MD simulations, being oriented in line with pre-TMH7 elbow helix, i.e., parallel to the lipid bilayer (Fig. 4d). This pre-TMH7 cholesterol hot spot was also observed for example in IF apo POPC:POPE:Chol (2:1:1) simulations (Fig. 4c). To a lesser extent, a second resolved cholesterol molecule remains in contact with the protein, by the pre-TMH1 elbow helix (Supplementary Fig. 33). Interestingly, this almost pseudo-symmetrical hotspot as compared to pre-TMH7 elbow helix was also observed in simulations carried out with IF *b*MRP1 conformations (Fig. 4d). Allosteric pathway analyses underlined the key role of cholesterol molecules close to pre-TMH1 and -TMH7 elbow helices in the information transduction from the substrate-binding pocket to NBSs as shown in Fig. 4e. Indeed, calculated betweenness of these cholesterol molecules clearly suggested that they actively participate in the allosteric communication from substrate binding site to NBSs. Finally, the last resolved cholesterol molecule observed at the interface between TMH5 and TMH8 does not stay in contact with protein core along MD simulations. However, calculated 2D density profiles of cholesterol suggest a mildly higher probability of cholesterol presence in this region, suggesting that the resolved molecule was exchanged along the MD simulation. Furthermore, such profiles also revealed higher density spots, such as the horizontally oriented cholesterol molecule by TMH4 (Supplementary Figs. 35–36).

## Discussion

*b*MRP1 is the only member of the ABC drug exporter C-family which has been resolved by cryo-EM so far, given that ABCC7/CFTR is a chloride channel[31] and sulfonylurea receptors (ABCC8 and ABCC9) are involved in the regulation of potassium channels[32]. Over the past decade, particular attention has been paid to MRP transporters including MRP1, MRP2 and MRP4 given their clinically and pharmacologically relevant roles in drug disposition as pointed out by the ITC[12,13]. In contrast to its cousin ABCB1/P-gp, knowledge about MRP1 dynamics and functions still remains fragmented although it has been resolved in multiple states, such as IF apo and substrate-bound states[8] and two OF states under pre-[4] and post-hydrolysis[5] conformations, respectively bound to either two ATP molecules or to ADP/ATP pair. In the present work, an extensive set of all-atom MD simulations were performed in order to capture conformational dynamics of *b*MRP1 in different states considering different mixtures of lipid bilayer models including PC and PE lipids as

well as cholesterol. We propose an MD-based computational approach in order to (i) complete the experimental observations made in detergents and (ii) highlight structural patterns which might be extended to, at least, other ABC C-family transporters.

Global conformational dynamics in POPC:POPE:Chol (2:1:1) shows significant variations of IF structures with respect to cryo-EM structures. In IF states, spontaneous closing of NBDs was systematically observed regardless of the presence of ATP molecules as pictured by IC angle and NBD distance values which all converged toward the same subspace. Regarding ABC conformational space, the presence of either ATP molecules in both NBSs or substrate in the TMD binding pocket mostly shifts the dynamics of NBD dimerisation by modulating the NBD twist value (Fig. 2b and Supplementary Fig. 5). Therefore, our MD simulations are in perfect agreement with recent observations[6,7] that wide-open IF structures may be unlikely in native membrane environments. Wide-open structures observed in cryo-EM experiments are thus believed to be due to artifacts owing to the use of non-physiological environments for structure resolution[7], in agreement with structural differences observed e.g., for P-gp reconstituted either in detergents or in nanodiscs[11]. The NBD1 13 amino-acid deletion, such as the absence of the "ball-and-socket" structure for NBD1, might weaken the coupling between NBD1 and TMH10/TMH11. In absence of substrate, this is associated with larger translational flexibility regarding NBD1 leading to the opening of NBD dimer. However, in presence of ATP molecules and substrate, only close NBD dimer conformations were observed (Fig. 2a) highlighting the allosteric communication between TMDs and both NBSs. Owing to the lower coupling between TMH10/TMH11 and NBD1, allostery between TMDs and NBDs is expected to be mostly mediated through NBD2. Interestingly, NBS1 was also formed even in absence of ATP molecules (Fig. 2a). This might also explain why IF-to-OF transitions were experimentally observed even in absence of ATP molecules[5]. However, such assumptions should be carefully considered given the recent resolution of an NBS degenerate ABC transporter adopting wide open IF conformation by means of cryo-EM using nanobodies[33]. Our results highlight the importance of NBD and NBS asymmetry as an evolution of ABC transporters which may result from energy saving while keeping transport function by requiring a single ATP hydrolysis.

MRP1 is expected to carry mostly anionic amphiphilic substrates contrary to P-gp which mostly transports hydrophobic substrates[2]. In agreement with structural observations, substrate access directly from either high- or low-density lipid tail regions of the membrane is very unlikely owing to the absence of an access channel in the lipid bilayer contrary to P-gp[8]. However, amphiphilic substrates might partition in the high-density polar head region. Therefore, MRP1 substrate access is expected to occur either directly from the cytoplasm or from the high-density polar head region, for which access might be possible via TMH4 and TMH6 (Supplementary Fig. 37). On the other side of *b*MRP1,

our MD simulations suggest that substrate access between TMH10 and TMH12 may be less likely for bulky substrates. However, such assumptions require further biochemical and structural investigations.

In the present work, we also investigated the interplay between lipid bilayer membranes and protein dynamics. First, it is important to note that present simulations were performed for few μs for each replica. Given the timescale of transport processes, present results can only be used to decipher lipid-protein interplay in the equilibrium regions of the different conformational states. By playing with different lipid bilayer models such as cholesterol-free and PE-free membranes, $b$MRP1 dynamics is modulated. In line with previous biophysical studies, the presence of cholesterol in PC-based membrane increased membrane stiffness which in turn reduced conformational dynamics of MRP1[29]. The impact of PE lipids onto local minimum structures is expected to be rather limited for IF states. However, interestingly, EC opening under OF conformation appeared more favourable in presence of PE (Fig. 4a). Globally, our results also suggest that MRP1 structure is likely to adopt similar structure for a given state regardless of the lipid bilayer composition. However, this may play a role in conformational transitions and kinetics. This hypothesis should be considered carefully for other MRP transporters. Indeed, in contrast to e.g., MRP2 and MRP4, MRP1 is a ubiquitous exporter which was observed in different cell types for which lipid bilayer composition might differ. We can thus infer that other MRP members might become more sensitive to lipid bilayer membrane compositions.

Out of the physical role of lipid bilayer on protein dynamics, our results support that lipid components also play an active role in transporter function. In line with computational observations made on other membrane proteins[11,26,34,35], lipid components are involved in the allostery from TMDs to NBDs. More importantly, cholesterol bind to pre-TMH1 and pre-TMH7 elbow helices are strongly involved in these allosteric pathways. Such findings pave the way regarding the role of lasso pre-TMD motif ($L_0$) which was shown to be required for the MRP1 transport function[8,17]. Even if the role of lipids might sound limited on MRP1 as compared to other membrane receptors and channels, more attention should be paid to lipid-protein interactions, not only regarding the biophysical impact on the protein dynamics but also as transport modulator as recently proposed for P-gp[11].

The present work provided structural insights into the function and the lipid-protein interplay of NBS degenerate ABCC transporters for further investigations such as the role of MRPs in local pharmacokinetics, including e.g., the impact of rare mutations/polymorphism as well as disease-based membrane lipid imbalance toward personalised medicine.

## Methods

**Construction of $b$MRP1 models embedded in lipid bilayer membranes.** In order to have an overview of milestone structures along the transport cycle of $b$MRP1, different conformations and bound states were considered in the present study: IF apo $b$MRP1, IF $b$MRP1-(ATP)$_2$, IF $b$MRP1-LTX, IF $b$MRP1-LTX-(ATP)$_2$ and OF $b$MRP1-(ATP)$_2$. The cryo-EM structures were used as starting structures for IF (PDB ID: 5UJ9[8] and 5UJA[8]) and OF conformations (PDB ID: 6BHU[4]). OF conformation was resolved using E1454Q mutant which was shown to lower the rate of ATP hydrolysis thus promoting OF structure determination[4]. This mutation was reverted manually for the present study. The so-called TMD$_0$ was not included in the present models as it has already been shown not to affect the substrate transport[3,4,8]. However, it was shown that the so-called pre-TMH1 lasso domain ($L_0$) is mandatory for MRP1 function while it was not totally resolved in any cryo-EM MRP1 structures[8,17]. Missing parts of $L_0$ domain was modelled using either I-Tasser (Iterative Threading ASSEmbly Refinement) server[36] or modeller v9.23[37] for IF and OF conformations, respectively. Indeed, I-Tasser initially failed to predict consistent $L_0$ domain for OF $b$MRP1-(ATP)$_2$ state as compared to IF model. Therefore, for sake of consistency, OF $b$MRP1-(ATP)$_2$ $L_0$ domain was built using modeller v9.23 based on the sequence but also IF $L_0$ domain model as template (Supplementary Table 13). To ensure the consistency between $L_0$ domain

models, structure and dynamics was monitored by assessing RMSF over MD simulations but also by comparing final $L_0$ domain model structures which converged toward similar secondary structures (Supplementary Figs. 38 and 39). Likewise, the missing loop between TMH6 and NBD1 was also added in the present models.

IF conformations were built in different bound states, namely apo, ATP$_2$-, LTX- and LTX-(ATP)$_2$-bound states while OF conformation was solely constructed in the ATP$_2$-bound state. IF $b$MRP1-(ATP)$_2$ and IF $b$MRP1-LTX-(ATP)$_2$ were constructed by superimposing separately NBD of 5UJ9 and 5UJA, respectively onto the NBDs of OF conformation in which both ATP molecules and Mg$^{2+}$ are bound to NBSs. All final models were shortly minimised in vacuum using the Amber18 package[38,39] to avoid unphysical steric clashes.

CHARMM-GUI input generator[40,41] was used to embed the different $b$MRP1 models into different lipid bilayers, namely pure POPC, POPC:Chol (3:1) and POPC:POPE:Chol (2:1:1) taking advantage of $b$MRP1 coordinates obtained from the OPM (Orientations of Proteins in Membranes) database[42]. IF apo $b$MRP1 and OF $b$MRP1-(ATP)$_2$ structures were also embedded into pure POPE and POPC:POPE (3:1) lipid bilayers in order to specifically investigate lipid-protein interactions with PE lipids. The resolved cryo-EM structure of OF $b$MRP1-(ATP)$_2$ also includes three resolved cholesterol molecules which were kept during all simulations in order to investigate their importance. From these different lipid bilayer compositions, POPC:POPE:Chol (2:1:1) mixture appeared the most relevant to model cell membranes. The other types of membranes were considered to help understand the role of each lipid in protein dynamics. The original total size of every system was *ca.* 120 × 120 × 180 Å$^3$ (see Supplementary Table 14 for system descriptions). To mimic physiological conditions, 0.15 M NaCl was used, and the systems were solvated using TIP3P explicit water molecules[43–45]. The final systems are made of *ca.* 245,000 atoms (see details in Supplementary Table 14).

**Molecular dynamics simulations.** CHARMM-GUI[40,41] outputs were converted to Amber format using AmberTools scripts[38,39] (namely, charmmlipid2amber.py and pdb4amber). Regarding ATP$_2$- and LTX-bound systems, substrate, nucleotides and Mg$^{2+}$ ions were added after building protein-lipid systems; therefore, neutrality was ensured by randomly removing the corresponding number of counterions. Amber FF14SB[46], Lipid17[47] and the modified DNA.OL15[48,49] force fields were used to respectively model protein residues, lipids and ATP molecules. Water molecules, Mg$^{2+}$ ions and counterions were modelled using the TIP3P water model[43–45] as well as the corresponding monovalent and divalent ion parameters from Joung and Cheatham[50,51]. LTX (Leukotriene C4) substrate parameters were derived from the Generalised Amber Force Field version 2 (GAFF2)[52] using the Antechamber software[53]. LTX partial atomic charges were derived from quantum mechanical based calculations at the HF/6-31G* level of theory, using the R.E.D. server[54]. Each system was simulated with periodic boundary conditions. The cutoff for non-bonded interactions was 10 Å for both Coulomb and van der Waals potentials. Long-range electrostatic interactions were computed using the particle mesh Ewald method[55].

Minimisation and thermalisation of the systems and MD simulations were carried out with Amber18 and Amber20 packages[38,39] using CPU and GPU PMEMD versions. Minimisation was carried out in four steps by sequentially minimising: (i) water O-atoms (20,000 steps); (ii) all bonds involving H-atoms (20,000 steps); (iii) water molecules and counterions (50,000 steps) and (iv) the whole system (50,000 steps). Each system was then thermalized in two steps: (i) water molecules were thermalized to 100 K during 50 ps under (N,V,T) ensemble conditions using a 0.5 fs time integration; (ii) the whole system was then thermalized from 100 K to 310 K during 500 ps under (N,P,T) ensemble conditions with 2 fs timestep in semi-isotropic conditions. Then, each system was equilibrated during 5 ns under (N,P,T) ensemble conditions with 2 fs timestep in semi-isotropic conditions, using Berendsen barostat. Production runs were then carried out at the microsecond scale with 2 fs integration timestep under (N,P,T) ensemble conditions with semi-isotropic scaling. Temperature was maintained using the Langevin dynamics thermostat[56] with 1.0 ps$^{-1}$ collision frequency. Constant pressure set at 1 bar was maintained with semi-isotropic pressure scaling using either Berendsen barostat[57] for IF apo $b$MRP1 and OF $b$MRP1-(ATP)$_2$ or Monte Carlo barostat for $b$MRP1-(ATP)$_2$, IF $b$MRP1-LTX and IF $b$MRP1-LTX-(ATP)$_2$. The latter was used to speed up computational time.

In order to ensure the ATP docking into NBS, restraint-MD simulations were carried out using a similar approach as proposed by Wen et al.[58]. Shortly, a set of distance-based restraints were applied between the A-loop tryptophan/tyrosine residues (Trp653 and Tyr1301, respectively for NBD1 and NBD2) and corresponding ATP purine moiety. Mg$^{2+}$-ATP-NBD arrangement was maintained by applying restraints between Mg$^{2+}$ ions and ATP phosphate groups as well as between Mg$^{2+}$ ions and Walker A serine and Q-loop glutamine residue (namely, Ser685 and Gln713 for NBD1 and Ser1333, Gln1374 for NBD2). Moreover, ATP phosphate moieties were also restrained with surrounding Walker A residues. All distances were restrained using harmonic potentials for which minimal distances and force constants are reported in Supplementary Tables 15–16. Distance-based restraints were applied for thermalisation and box equilibration steps. They were then smoothly removed along the first 10 ns of production runs. Restraints for Mg$^{2+}$ ions were kept during the whole simulation.

Snapshots were saved every 100 ps. For each system, three replicas were performed to better sample the local conformational space. Each production run

was carried out for 2.0–2.5 µs and 1.5–2.0 µs, respectively for IF and OF models (Supplementary Table 17). Indeed, simulations performed using OF model reached the equilibrium faster than IF conformations (see time-dependent RMSDs in Supplementary Fig. 40). In the present study, the aggregated total MD time is 112.4 µs.

**Analysis and visualisation.** Simulations were analysed using the CPPTRAJ[59] package, and in-house Python scripts taking advantage of MDAnalysis module[60,61]. Plots were obtained using the matplotlib v3.3.1 Python package[62]. Structure visualisation and rendering were prepared using VMD software[63] (v1.9.3 and the alpha-v1.9.4). The so-called ABC structural parameters (i.e., IC angle, EC angle, NDB distance, NBD twist and EC distance) were calculated using the same definition as proposed by Hofmann et al.[1] for IC angle, EC angle, EC distance and NBD distance or Moradi et al.[18] for NBD twist. Shortly, IC angle describes the IC opening of substrate entry and is defined by the angle between two vectors; both starting from the centre-of-mass of the whole extracellular region and directed toward either the IC region of TMH1, TMH2, TMH3, TMH6, TMH10 and TMH11 or the IC region of TMH4, TMH5, TMH7, TMH8, TMH9 and TMH12. Likewise, EC angle describes the EC opening for substrate release and is defined by the angle between two vectors starting from the centre-of-mass of both NBDs and directed toward either the EC region of TMH1, TMH2, TMH9, TMH10, TMH11, and TMH12 or the EC region of TMH3, TMH4, TMH5, TMH6, TMH7, and TMH8. EC distance was defined as the distance between the EC regions of TMH1, TMH2, TMH9, TMH10, TMH11, and TMH12 and the EC region of TMH3, TMH4, TMH5, TMH6, TMH7, and TMH8. NBD distance was defined as the distance between the two NBD centres-of-mass. Since these definitions were applied to the present MD models of bMRP1 but also to available resolved structures of other ABC transporters, intracellular and extracellular regions were defined based on the membrane thickness proposed in the OPM database[42]. Residue selections for each system and each structure parameter are reported in Supplementary Table 1. For each system and each lipid bilayer, local free energy landscape was calculated using the InfleCS approach which takes advantage of Gaussian Mixture Models (GMM)[23,64]. Structural parameters (i.e., IC and EC angles, NBD distance and twist) were taken to monitor the free energy landscape using a grid size set at 80, from 2 to 12 gaussian components for each GMM obtained by a maximum of 20 iterations. The relevance of InfleCS approach strongly relies on the quality of sampling during MD simulations. In the present work, InfleCS only pictures the free energy landscape around the local minima sampled during our MD simulations. Furthermore, MD sampling and relevance of InfleCS for the present systems were ensured by calculating the convergence profiles for each structural parameter separately (Supplementary Figs. 41–43).

The water pore profile of bMRP1 TM cavity was calculated using the Hole2.2 software[65] by taking snapshots every 10 ns. Average TM pore profile was calculated by averaging the equilibrated part of the trajectories considering all replicas. Time-dependent TM pore evolution from the beginning was also calculated at selected depths in lipid bilayer membrane (z = 18, 5, −15 and −22 Å) by bootstrapping every 100 ns along MD trajectories (i.e. 10 snapshots × 3 replica each 100 ns). Profiles were then averaged and PC P atom z-density profiles were used to define the centre of the lipid bilayer membrane (z = 0). Regarding tilt angles, all systems were aligned to the OF bMRP-(ATP)$_2$ model embedded in POPC:POPE:Chol (2:1:1). Lipid distributions were obtained using the CPPTRAJ[59] package. Lipid occupancies were calculated for each polar head (i.e., PE or PC) or whole cholesterol molecule around the protein, considering threshold occupancies at 50 or 80%. Leaflet dependent lipid densities were calculated using the grid keyword in CPPTRAJ focusing on polar head lipids (PE or PC) or cholesterol OH-group. Focus was paid only to the high-density polar head region obtained by monitoring z-dependent density peaks of phosphate atoms for PC and PE lipids or O-atom for cholesterol molecules. Membrane thicknesses were obtained using the MEMBPLUGIN for VMD[63,66]. Free energy deformation was assessed by calculating the free energy cost of membrane deformation in presence of bMRP1 as well as the free energy references for flat membranes, using the CTMDapp software[30]. All parameters used for these calculations are reported in Supplementary Table 18. Given the present sampling, the size of the protein and the relevance of bending and compressibility moduli in present all-atom simulations, results shown in Fig. 4c are discussed qualitatively.

Principal component analyses were also performed using the CPPTRAJ[59] package, focusing on the ABC core, defined by backbone atoms of TMH1 to TMH12, NBD1 and NBD2. System variabilities were investigated by carrying out independent PCA for each system for which each replica was aligned to an average structure of the system. Network analyses were performed using the VMD Network Analysis plugin[63,67]. Dynamic cross-correlation matrices (DCCM) were calculated separately for each replica on which Cα-atoms were selected as nodes and all the default restrictions (notSameResidue, notNeighboringCAlpha, notNeighboringPhosphate, notNeighboringResidue) were applied. Communities were then calculated using gncommunities[67]. Allostery network pathways were determined using the recent Allopath approach developed by Westerlund et al.[26,27]. Shortly, the distant "communication efficiency" between two domains was obtained from the contact map and the mutual information matrix of protein residues as well as non-protein interactors such as surrounding lipid and bound molecules (e.g., nucleotides and substrate). Such an approach also provides a betweenness profile which pictures the involvement of each component (i.e., residue, lipid, substrate and nucleotide). For each lipid molecule, three nodes were defined corresponding to the polar head group and the two lipid tails (Supplementary Fig. 44). ATP was also split into three nodes: purine and ribose moiety as well as triphosphate tail. LTX substrate was divided into three nodes: glutathione moiety, poly-unsaturated tail and hydroxypentanoic acid. Mg$^{2+}$ ions and cholesterol molecules were considered as one node each. Atom selections per node are reported in Supplementary Fig. 44. Allosteric pathways were calculated from substrate-binding pocket to each NBS, separately, as defined in Supplementary Table 9. H-bond analyses (Supplementary Figs. 45 and 46) were performed using CPPTRAJ[59] in which distance and angle cutoffs were set at 3.5 Å and 120°, respectively. Calculated DCCM using CPPTRAJ are also available in Supplementary Figs. 47–50. Structural descriptions of local minima were performed only on the equilibrated part of the MD simulations, i.e., considering the last 800 ns as shown from RMSD profiles reported Supplementary Fig. 41.

**Statistics and reproducibility.** For all MD analyses described in this paper, data were derived from n = 3 MD simulations for each state (i.e., IF apo bMRP1, bMRP1-LTX, bMRP1-(ATP)$_2$, bMRP1-LTX-(ATP)2 and OF bMRP1-(ATP)$_2$) and lipid bilayer membranes (i.e., pure POPC, POPC:Chol (3:1), and POPC:-POPE:Chol (2:1:1) for all states but also pure POPE and POPC:POPE (3:1) for IF apo bMRP1 and OF bMRP1-(ATP)$_2$). In other words, 19 systems were considered, for 57 simulations. Analyses were systematically performed considering each system independent. Averages and standard deviations were generally obtained by pulling data from each replica altogether to display the sampling conformational variability. For H-bond analyses, free energy deformation and allosteric communications, results were calculated for each replica separately and average and standard deviations and errors were obtained over the three replicas treated independently.

**Reporting summary.** Further information on research design is available in the Nature Portfolio Reporting Summary linked to this article.

## Data availability

Dataset generated from MD simulations conducted in this work are available from the authors on reasonable requests. Source data underlying results and plot presented in the main figures, MD inputs as well as initial and final coordinate configurations for each state and lipid model, and Supplementary Movies 1–5 can be downloaded at the following Zenodo access link: https://doi.org/10.5281/zenodo.7541178.

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

## Acknowledgements

The authors thank Dr. Benjamin Chantemargue and Dr. Mehdi Benmameri for stimulating and fruitful scientific discussions. We thank Xavier Montagutelli from the IT support of Limoges University for technical support regarding supercomputer facilities. Calculations were performed using CALI ("CAlcul en LImousin") and "Baba Yaga" supercomputers hosted by Limoges University, as well as on the "Jean-Zay" national supercomputer from IDRIS HPC resources, under the allocations 2020-A0080711487 and 2021-A0100711487 made by GENCI. This work was supported by grants from the "Agence Nationale de la Recherche" (ANR-19-CE17-0020-01 IMOTEP and ANR-21-CE18-0030 RAPRACLID), Région Nouvelle Aquitaine and "Institut National de la Santé et de la Recherche Médicale" (INSERM, AAP-NA-2019-VICTOR).

## Author contributions

Á.T. and F.D.M. conceived the study. Á.T. and F.D.M. conducted all the simulations. Á.T., A.J., V.C. and F.D.M. analysed the simulations. Á.T. and F.D.M. interpreted the

results and discussed them together with A.J. and V.C. Á.T. and F.D.M. wrote the manuscript and it was edited, reviewed and approved by all authors.

## Competing interests

The authors declare no competing interests.
