## [Peer Review File · Communications Biology]

Reviewers' comments:

Reviewer #1 (Remarks to the Author):

Manuscript by Tóth et al reports on a microsecond-level molecular dynamics study of MRP1, an NBS degenerate ABC transporter. They suggest that the asymmetric NBD behavior is ensured by lower signal transduction from NBD1 to the rest of the protein owing to the absence of ball-and-socket conformation between NBD1 and coupling helices. They are also hypothesizing that lipid composition has a limited impact on the mechanism, mostly by affecting transport kinetics. While this is a significant study involving an extensive set of MD simulations, there are major concerns that need to be addressed regarding the soundness of the methodology and the presentation of the results:

1. The authors only use POPE, POPC, and Chol and the simulations although long with the current standards of the field but are still short given the timescales involved in the transport process. I think the conclusion that the lipid composition has a minor effect on the mechanism is an overstatement. The lipid-dependent behavior particularly becomes important when in vitro experiments use non-native lipids or detergents to only mimic the environment of the protein. It would have been, perhaps, more useful to compare the behavior of the protein in an environment similar to that used in the cryo-EM experiment and compare that to the one presented here.

2. Why two different servers are used for modeling the missing parts of L0 domain? It is important to stay consistent, otherwise, it is difficult to explain the differences observed.

3. The resolved cholesterol molecules are kept in all simulations. Do they stay or leave the binding site?

4. The authors state "The so-called ABC structural parameters (i.e., IC angle, EC angle, NDB distance, NBD twist and EC distance) were calculated using the same definition as proposed by Hofmann et al". First, there is a typo (NDB) and more importantly, the Hoffman et al does not include NBD twist as far as I understand. Is there any other reference for it or the NBD twist is what the authors defining here for the first time?

5. The InfleCS method used for free energy estimation has recently been developed and tested on toy models; however, this method is not an enhanced sampling technique. Therefore, for a large system like a transporter it suffers from insufficient sampling. Also there is no information on the convergence of the free energy calculations.

Reviewer #2 (Remarks to the Author):

The author used all-atom long time scaled MD simulations to study the lipid property of an ABC transporter. It yielded a total of 110 us MD simulations. This work provided a useful information about how lipids behaved around membrane proteins and provided an insightful view towards basic research and fundamental questions.

Since there are many properties of lipids can be studied as well as the protein structure itself. I would suggest the following important properties should be included for analysis as well:

(1) the diameter size of TM pore along the whole MD simulations. How it changes during such long MD simulations.

(2) the stability of the transporter. Is the simulation and protein stable during such long simulations?

(3) since the author introduced hetrogenous lipids. What is the average density of each lipid component during the last 100 ns?

(4) What is the deformations energy around lipid-protein interface? One can take a look at the following paper :

Quantitative Modeling of Membrane Deformations by Multihelical Membrane Proteins: Application to G-Protein Coupled Receptors (2011) Biophysical Journal

Reviewer #1 (Remarks to the Author):

Manuscript by Tóth et al reports on a microsecond-level molecular dynamics study of MRP1, an NBS degenerate ABC transporter. They suggest that the asymmetric NBD behavior is ensured by lower signal transduction from NBD1 to the rest of the protein owing to the absence of ball-and-socket conformation between NBD1 and coupling helices. They are also hypothesizing that lipid composition has a limited impact on the mechanism, mostly by affecting transport kinetics. While this is a significant study involving an extensive set of MD simulations, there are major concerns that need to be addressed regarding the soundness of the methodology and the presentation of the results:

We thank the reviewer for the fair and constructive feedbacks. We hopefully have addressed most of his/her comments, which clearly helped improve the present study.

Comment 1. The authors only use POPE, POPC, and Chol and the simulations although long with the current standards of the field but are still short given the timescales involved in the transport process. I think the conclusion that the lipid composition has a minor effect on the mechanism is an overstatement. The lipid-dependent behavior particularly becomes important when *in vitro* experiments use non-native lipids or detergents to only mimic the environment of the protein. It would have been, perhaps, more useful to compare the behavior of the protein in an environment similar to that used in the cryo-EM experiment and compare that to the one presented here.

We would like to sincerely thank the reviewer for this fair comment. Indeed, we overstated the “minimal” impact of lipid bilayer membrane in the discussion. It is worth mentioning that we were aware of the MD timescale limitation and thus, conformational sampling as stated e.g., (p18, line 25-28):

“Even if these calculations suggest a relatively limited overall impact of membrane composition while comparing bMRP1 structures in different lipid bilayer models, the sampling is not sufficient to catch the role of lipids over the transport cycle (Supplementary Fig. 1-6 and 9).”

We actually mostly wanted to stress out that overall structures of the different bMRP1 state subspace are similar regardless of the lipid bilayer composition. We anyway agree that the sampling is still short as compared to the overall transport process, as we tried to underline more along the manuscript. This is true that the initial version of the present manuscript overstated the “minor effect” of lipid bilayer composition. Therefore, we smoothed both the “results” and “discussion” sections. For instance, sentences were added or rephrased in order to stress (i) limitations and (ii) what we observed directly from MD simulations, rather than over-interpretation, e.g., p25, line 8-12:

“First, it is important to note that present simulations were performed for few ms for each replica. Given the timescale of transport processes, present results can only be used to decipher lipid-protein interplay in the equilibrium subspace regions of the different conformational states.”

or p19, line 23-26:

“This suggests that even though lipid composition seems to have a rather limited impact while comparing structures of bMRP1 local minima in different lipid bilayer

membranes, lipid composition is expected to affect conformational transitions, and thus, in turn, play a role in the kinetics of substrate transport by bMRP1.“

We also perfectly agreed with the idea to compare MD simulations performed in similar environment to that used in the Cryo-EM with our symmetric lipid bilayer models. However, after the careful checking on available parameters in the AMBER force-field family, we unfortunately were not able to carry out such simulations. Indeed, AMBER force fields were used for all simulations performed in the present manuscript; however, parametrization of detergents similar to those used in experiments would have been too risky, in our opinion. As an alternative, we also considered using CHARMM-based force field family for detergent simulations, but comparison of results from simulations performed with two different FF families may also lead to comparison misinterpretation.

It is worth mentioning that our assumption that detergents may lead to more open IF structures was supported by observations and interpretation in the literature (e.g., *Chem. Sci.* **12**, 6293–6306 (2021), *FEBS Lett.* **594**, 3815–3838 (2020)). However, in the present context, it is important to note that we also smoothed the statement about the low probability of wide open IF conformation, accounting the recent observation made on wide open IF conformation for MsbA ABC transporter (*Sci. Adv.* **8**, eabn6845 (2022)), (see p.24 line 1-6):

“Wide open structures observed in cryo-EM experiments were thus believed to be due to artifacts owing to the use of non-physiological environments for structure resolution⁷, in agreement with structural differences observed e.g., for P-gp reconstituted either in detergents or in nanodiscs¹¹. However, such assumptions should be carefully considered given the recent resolution of an NBS degenerate ABC transporter adopting wide open IF conformation by means of Cryo-EM using nanobodies³⁴.”

Comment 2. Why two different servers are used for modelling the missing parts of L₀ domain? It is important to stay consistent, otherwise, it is difficult to explain the differences observed.

We thank the reviewer for pointing out the lack of clarity in our model construction. We thus clarify the different steps in the “Methods” section. We had to model the L₀-domain for OF conformation using Modeller v9.23. By using IF conformation as a template without accounting the positional difference of surrounding TMHs, I-TASSER failed at proposing consistent L₀ structure. Thus, we decided to model OF L₀-domain partially based on the sequence but also accounting the secondary structure of IF L₀-domain to ensure the consistency between these states (see details shown in the following table).

269-RKQPVKIV-276	L ₀ of IF system
277-YSSKDKPAKPKGSSKVDV-293	sequence
294-NEEAEALIVKCPQKERD-310	L ₀ of IF system

To ensure the reliability of our L₀-domain model, its root-mean square fluctuation was monitored showing the expected high flexibility, especially for the loop not included in the initial IF cryo-EM structures (around 80-95, assuming L₀ numbering starting at 1, see RMSF plot below). Finally, to confirm the consistency between the different L₀-constructions, we have checked that L₀-domain converged towards similar structure as shown in the figure below. For sake of clarity and readability, more details were

included in the present version of the manuscript. The following plots are also now included in Supplementary information (see Page 27, Line 13-21, Supplementary Figures 38-39 and Supplementary Table 13):

“Missing parts of L_0 domain was modelled using either I-Tasser (Iterative Threading ASSEmbly Refinement) server³⁷ or modeller v9.23³⁸ for IF and OF conformations, respectively. Indeed, I-Tasser initially failed to predict consistent L_0 domain for OF bMRP1-(ATP)₂ state as compared to IF model. Therefore, for sake of consistency, OF bMRP1-(ATP)₂ L_0 domain was built using modeller v9.23 based on the sequence but also IF L_0 domain model as template (Supplementary Table 13). To ensure the consistency between L_0 domain models, structure and dynamics was monitored by assessing RMSF over MD simulations but also by comparing final L_0 domain model structures which converged toward similar secondary structures (Supplementary Fig. 38-39).”

Root-mean square fluctuation of L_0 . The part which was modelled using the sequence and not the loop from the IF model (around 80-95) is the most flexible part in the IF models, as well.

a) Starting frames converge to the same conformation shown by **b)** the final frames. Top view., aligned on TH1-3,6,10-11 (bundle A and C). IF apo is blue, IF ATP-bound ochre, IF LTX-bound yellow, IF LTX-ATP-bound green, and OF ATP-bound mauve/pink.

Comment 3. The resolved cholesterol molecules are kept in all simulations. Do they stay or leave the binding site?

Indeed, this aspect was not sufficiently discussed in the original version of the manuscript. As shown on Figure 4d-e, cholesterol occupancies were calculated over MD simulations exhibiting cholesterol binding hotspots. Among them, MD simulations suggests that one is particularly importance since it is conserved all along the MD simulations. To a lesser extent, on the other side of the protein, a second resolved cholesterol molecule remains close to pre-TMH1 elbow helix. Interestingly, as stated in the discussion section, both cholesterol molecules seem to play a role in the allosteric communication between substrate binding pocket and NBSs. Finally, the last resolved cholesterol molecule at the interface between TMH5 and TMH8 “left” his initial position. However, from lipid density analyses suggested by reviewer #2, we observed that the TMH5/8 interface exhibits a mildly favoured region for protein-cholesterol interaction, suggested that the initial resolved cholesterol has been exchanged along MD simulations.

It has been clarified in the manuscript p. 22, line 8-26 and p.23 line 1-2:

“Electron density maps revealed three cholesterol molecules bound to the resolved OF bMRP1 structure from which two was maintained near to its initial position with a probability higher than 50% (Supplementary Fig. 33). Interestingly, one by the pre-TMH7 is strongly (more than 80%) maintained along the MD simulations, being oriented in line with pre-TMH7 elbow helix, i.e., parallel to the lipid bilayer (Fig. 4d). This pre-TMH7 cholesterol hot spot was also observed for example in IF apo POPC:POPE:Chol (2:1:1) simulations (Fig. 4c). To a lesser extent, a second resolved cholesterol molecule remains in contact with the protein, by the pre-TMH1 elbow helix

(Supplementary Fig. 33). Interestingly, this almost pseudo-symmetrical hotspot as compared to pre-TMH7 elbow helix was also observed in simulations carried out with IF bMRP1 conformations (Fig. 4d). Allosteric pathway analyses underlined the key role of cholesterol molecules close to pre-TMH1 and -TMH7 elbow helices in the information transduction from the substrate-binding pocket to NBSs as shown in Fig. 4e. Indeed, calculated betweenness of these cholesterol molecules clearly suggested that they actively participate in the allosteric communication from substrate binding site to NBSs. Finally, the last resolved cholesterol molecule observed at the interface between TMH5 and TMH8 does not stay in contact with protein core along MD simulations. However, calculated 2D density profiles of cholesterol suggest a mildly higher probability of cholesterol presence in this region, suggesting that the resolved molecule was exchanged along the MD simulation. Furthermore, such profiles also revealed higher density spots, such as the horizontally oriented cholesterol molecule by TMH4 (Supplementary Fig. 35-36).”

Furthermore, 50% occupancy for resolved cholesterol molecules with respect to protein core was also reported in new Supplementary Figure 33:

Calculated binding hotspots obtained from cholesterol defined by presence likelihood higher than 50%. Cryo-EM resolved cholesterols are coloured violet, other cholesterols green.

4. The authors state "The so-called ABC structural parameters (i.e., IC angle, EC angle, NDB distance, NDB twist and EC distance) were calculated using the same definition as proposed by Hofmann et al". First, there is a typo (NDB) and more importantly, the Hoffman et al does not include NDB twist as fas as I understand. Is there any other reference for it or the NDB twist is what the authors defining here for the first time?

We agree that our initial description of ABC structural parameter was misleading. It is worth mentioning that such structural parameters were first proposed by Moradi et al. (Moradi, M. & Tajkhorshid, E. Mechanistic picture for conformational transition of a membrane transporter at atomic resolution. Proc. Natl. Acad. Sci. 110, 18916–18921

(2013)). Hofmann et al. used these parameters to compare many resolved structures, except NBD twist. In the present study, we decided to include all of them, since NBD distance is, at least, partially correlated to IC angle.

We thus stressed out that several studies were considered to choose ABC structural parameters, p. 5, lines 24-25 and p.6 line 1-4:

“To examine the conformational space sampled during the simulations in POPC:POPE:Chol (2:1:1) accounting for bound states of bMRP1, different structural descriptors were considered according to previous studies^{1,11,18}. Namely, intracellular (IC) and extracellular (EC) angles were monitored for TMDs while NBD distance and NBD rocking-twist angle were used for NBDs (Fig. 1 and Supplementary Fig. 1-6); the latter being known to adapt along OF-to-IF transition in ABCB1/P-gp¹⁸. »

And in the Method section (p. 30 lines 19-21) :

“The so-called ABC structural parameters (i.e., IC angle, EC angle, NBD distance, NBD twist and EC distance) were calculated using the same definition as proposed by Hofmann et al.¹ for IC angle, EC angle, EC distance and NBD distance or Moradi et al. for NBD twist¹⁸.”

5. The InfleCS method used for free energy estimation has recently been developed and tested on toy models; however, this method is not an enhanced sampling technique. Therefore, for a large system like a transporter it suffers from insufficient sampling. Also there is no information on the convergence of the free energy calculations.

We sincerely thank the reviewer for this comment of particular importance regarding the use of InfleCS method. To ensure its present reliability, we calculated convergence for the free energy calculations exhibiting acceptable convergence given chemical accuracy of the methods, (p. 31, lines 14-19 and p.):

“The relevance of InfleCS approach strongly relies on the quality of sampling during MD simulations. In the present work, InfleCS only pictures the free energy landscape around the local minima sampled during our MD simulations. Furthermore, MD sampling and relevance of InfleCS for the present systems were ensured by calculating the convergence profiles for each structural parameter separately (Supplementary Fig. 41-43).”

and

“The relevance of the InfleCS was assessed by monitoring the convergence of free energy profiles along MD simulations (see Methods section) suggesting an acceptable sampling of the local subspace.”

The following Supplementary Figure 41-43 are now included in the ESI. In line with reviewer's comments, we also clarified in the method section to underline that the present use of InfleCS can only provide information about the sampling around the local minima, around our sampling. We are confident that observations made from present MD simulations (e.g., using μ s-scaled MD simulations for 3 replicas) are sufficient to document structural differences between states, as shown by InfleCS convergence profiles as well as the evolution of e.g., time-dependent RMSD or ABC structural parameters along MD simulations.

POPC:POPE:Chol (2:1:1)

POPC:Chol (3:1)

Pure POPC

Reviewer #2 (Remarks to the Author):

The author used all-atom long time scaled MD simulations to study the lipid property of an ABC transporter. It yielded a total of 110 us MD simulations. This work provided a useful information about how lipids behaved around membrane proteins and provided an insightful view towards basic research and fundamental questions.

Since there are many properties of lipids can be studied as well as the protein structure itself. I would suggest the following important properties should be included for analysis as well:

Comment 1. the diameter size of TM pore along the whole MD simulations. How it changes during such long MD simulations.

We thank the reviewer for his/her fruitful comment. We performed such analyses revealing interesting differences that are described p. 8 lines 26-29 and p.9 lines 1-7 as well as p.18 lines 22-25 :

“On the other hand, our calculations suggest that only EC angle is affected by lipid bilayer composition but only for OF conformations. Likewise, it is important to note that calculated cavity radii (Supplementary Fig. 9) exhibited small differences while comparing lipid bilayer compositions.”

Furthermore, (time-dependent) pore radius profiles are also now included as Supplementary Figure S9 to 13 (see below).

Average z-dependent transmembrane pore radii.

Time-dependent pore radius profiles at $z = 18 \text{ \AA}$

Time-dependent pore radius profiles at $z = 5 \text{ \AA}$

Time-dependent pore radius profiles at $z = -15 \text{ \AA}$

Time-dependent pore radius profiles at $z = -22 \text{ \AA}$

Comment 2. the stability of the transporter. Is the simulation and protein stable during such long simulations?

Time-dependent root-mean square deviation (RMSD) profiles (Supplementary Fig. 40), revealed stable systems along MD simulations. Interestingly, we also observed that the simulation time required for reaching protein stability was longer in POPC than in mixture, as originally pictured by ABC structural parameters but also from RMSD

profiles (Supplementary Fig. 40). These RMSD-based stabilities are now underlined in Methods section in which the time used for analyses is now explicitly reported:

“Structural descriptions of local minima were performed only on equilibrated part of the MD simulations, i.e., considering the last 800 ns as shown from RMSD profiles reported Supplementary Fig. 40.”

Comment 3. since the author introduced hetrogenous lipids. What is the average density of each lipid component during the last 100 ns?

We sincerely thank the reviewer for his/her fruitful suggestions. In line with former comment with Reviewer #1, we are convinced that such analyses were useful to support our initial investigations. We therefore calculated 2D density profiles for each lipid types and cholesterol (Supplementary Figures 34-36), Such results being complementary to those pictured in Figure 4.

We considered each leaflet separately to picture the asymmetric protein-lipid interplay as details in the “Methods” section (p31, line 4-5).

Cholesterol in POPC:Chol (3:1)

Cholesterol in POPC:POPE:Chol (2:1:1)

PE-lipid in POPC:POPE:Chol (2:1:1)

Comment 4. What is the deformations energy around lipid-protein interface? One can take a look at the following paper:
Quantitative Modeling of Membrane Deformations by Multihelical Membrane Proteins: Application to G-Protein Coupled Receptors (2011) Biophysical Journal

We are really grateful to the reviewer for this very interesting and exciting suggestion. Following his/her advice, we assessed the membrane deformation free energy at the lipid-protein interface using the approach suggested above. It is now detailed in the Methods section for which parameters used for these calculations are also reported.

It is important to note that such calculations were relatively time consuming, and the technical framework was challenging given that the discrepancies between JAVA software requirements and OS development since 2011. It was performed for POPC, POPC:Chol (3:1) and POPC:POPE:Chol (2:1:1) lipid bilayer models. Detailed results are now reported in Supplementary Table 10-12 while an overview of conformation-dependent free energy deformation is now included in main Figure 4 (including a new panel c). From the qualitative point of view, we clearly see that the strong interplay between lipids and protein. Interestingly, MRP1 tends to destabilize POPC and, to a lesser extent POPC:Chol (3:1) lipid bilayer. In contrast, in the model closer to realistic membrane, MRP1 seems to stabilise lipid bilayer membrane (p. 21, lines 22-29 and p. 22 lines 1-3):

“Membrane free energy deformations³¹ were also assessed to document on the impact of bMRP1 conformations onto lipid bilayer structures (Supplementary Tables 10-12 and Fig. 4c). While calculations suggest that the presence of bMRP1 destabilise pure POPC lipid bilayer structure, the opposite trend was interestingly observed in POPC:POPE:Chol (2:1:1) in which membrane is stabilized in presence of bMRP1 ($DG_{\text{deformation}} < 0$, see Fig. 4c). An intermediate behaviour was observed in POPC:Chol (3:1) lipid bilayer membrane for which the presence of bMRP1 globally destabilized lipid bilayer structure but to a significantly lesser extent than in pure POPC. Except for simulations performed in pure POPC lipid bilayer, calculated deformation free energies are larger for IF apo bMRP1 state than for other conformations, likely due to its aforementioned larger flexibility in absence of ATP and/or substrate.”

New Figure 4.

In addition, some typos were corrected:

In the ESI

- Supplementary Figures 1-8 and 30 as the limit of the x axis was not well handled.
- Supplementary Movies 1-5 as the colouring was not in agreement with the Figure 1. colouring.
- INSERM to Inserm

In the main text

- Figure 1b-c x axis
- Figure 2a representative snapshots
- Figure3b colouring
- INSERM to Inserm

REVIEWERS' COMMENTS:

Reviewer #1 (Remarks to the Author):

The authors have addressed all the comments previously raised and the manuscript presents a scientifically sound and interesting story.

Reviewer #2 (Remarks to the Author):

All my concerns were resolved. Thus I would recommend for publication .